WIND ENERGY SCIENCE DISCUSSIONS

# Effect of Rotor Design on Energy Performance and Cost of Stationary Unmoored Floating Offshore Wind Turbines

Aurélien Babarit[1], Maximilien André[1], and Vincent Leroy[1]

[1]Nantes Université, Centrale Nantes, CNRS, LHEEA, UMR6598, 1 rue de la Noe, 44300 Nantes, France

**Correspondence:** Aurélien Babarit (aurelien.babarit@ec-nantes.fr)

**Abstract.** This paper investigates the effect of rotor design on energy performance and cost of a Stationary Unmoored Floating Offshore Wind turbine (SUFOWT). A SUFOWT is a Floating Offshore Wind Turbine (FOWT) for which a dynamic positioning (DP) system is used in lieu of a mooring system for station-keeping. It is particularly well suited for deployment in the far-offshore.

Previous studies have shown that positive net power production can be achieved with SUFOWTs depending on number and size of thrusters, and wind turbine characteristics. However, they did not consider rotor design. This gap is addressed in the present paper. The study is based on a physical engineering model. The wind turbine rotor design is represented by its rated induction factor.

Results show that the optimal rated induction factor is smaller than the usual value of 1/3 both from the perspective of energy performance and cost of energy. Thus, wind turbine rotors designed for SUFOWTs should be developed to optimize their cost. However, results show that the cost of energy reduction is somehow limited, of the order of 2.5 to 4.3% for the considered designs.

## 1 Introduction

Unmoored Floating Offshore Wind Turbines (UFOWTs) have been proposed as alternatives to conventional Floating Offshore Wind Turbines (FOWTs) (Tsujimoto et al., 2009; Xu et al., 2021) or as a solution to harness far-offshore wind energy (which is inaccessible for conventional FOWTs) (Alwan et al., 2021; Raisanen et al., 2022; Annan et al., 2023; Santarromana et al., 2024). The key difference between conventional FOWTs and UFOWTs is that UFOWTs use thrusters in lieu of mooring systems for position control. In addition, UFOWTs may be equipped with on-board energy storage systems (e.g. batteries, hydrogen, etc.) to avoid grid-connection. Examples of UFOWTs conceptual designs are shown in Figure 1.

UFOWTs may further be classified as Mobile Unmoored Floating Offshore Wind Turbines (MUFOWTs) or Stationary Unmoored Floating Offshore Wind Turbines (SUFOWTs) depending whether they are allowed to move in the ocean space. Note that in the case of SUFOWTs, the thrusters and control system is *de facto* a Dynamic Positioning (DP) system.

Obviously, the energy consumed by the thrusters must be smaller than the energy generated by the wind turbine for the UFOWT concept to make sense. Fortunately, previous studies have shown that it can be the case both for MUFOWTs and SUFOWTs. In (Xu et al., 2021), a SUFOWT design consisting of the 5 MW NREL wind turbine (Jonkman et al., 2009)

WIND ENERGY SCIENCE DISCUSSIONS

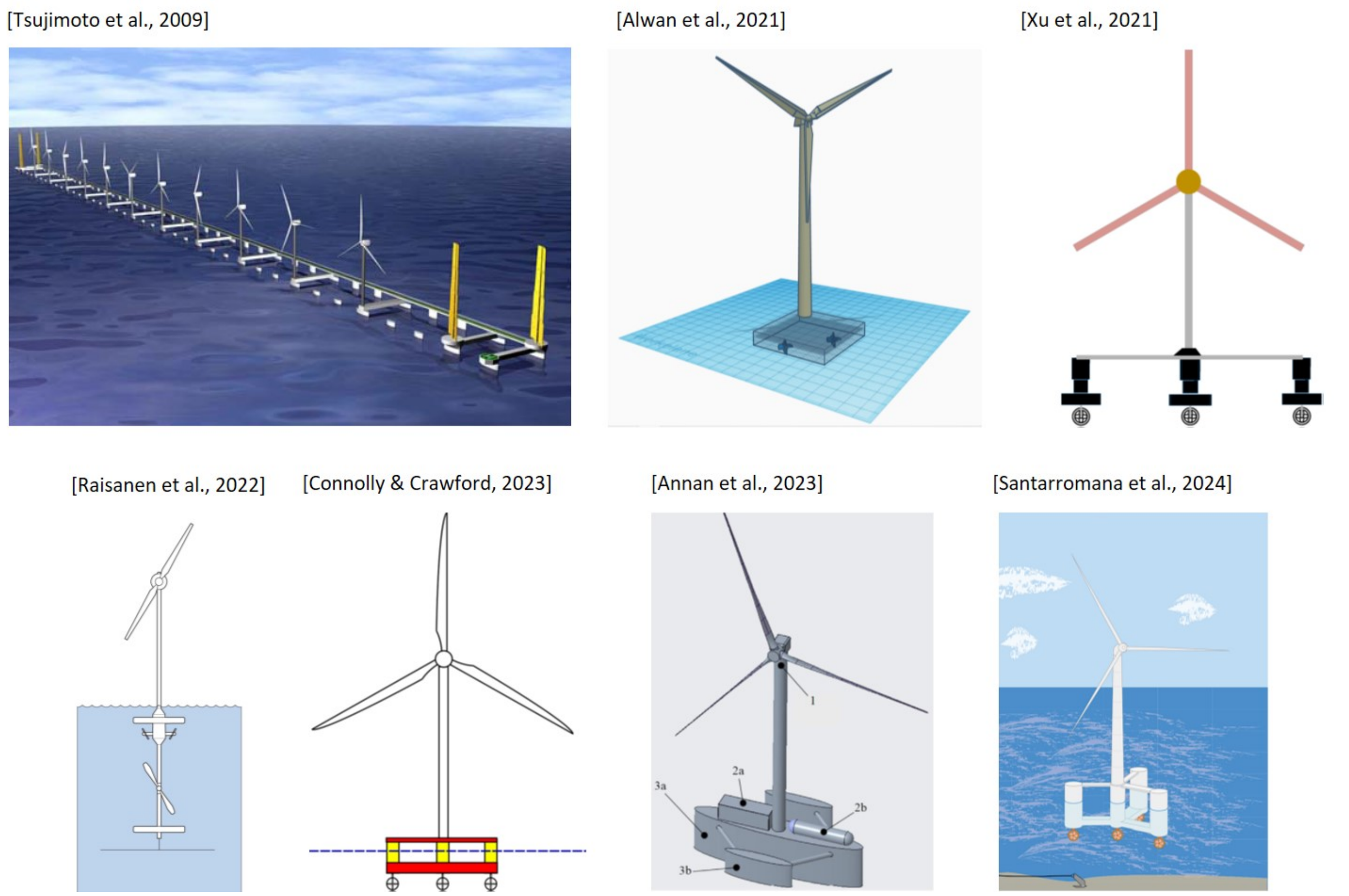

**Figure 1.** Examples of UFOWTs conceptual designs

mounted on a four-columns semi-submersible floater equipped with four 3.6 m diameter thrusters was considered. The thrusters power was found to be slightly less than 50% of the power generated by the wind turbine at rated wind speed. Moreover, the power ratio (ratio of thrusters power consumption to wind turbine generated power) was shown to decrease with increasing thrusters diameter and decreasing wind turbine diameter. Using the same SUFOWT design, Connolly and Crawford (2023) found that rated wind speed is the worst-case for the power ratio, because in region II (*i.e.* from cut-in wind speed to rated wind speed), the aerodynamic thrust increases with the square of the wind speed. In contrast, in region III (between rated wind speed and cut-out wind speed), Connolly and Crawford (2023) found that the power ratio decreases with increasing wind speed, due to the aerodynamic thrust decreasing with increasing wind speed (resulting in the thrusters consuming less power) while the generated power is constant. Alwan et al. (2021) considered a SUFOWT design consisting of a 2 MW wind turbine mounted on a square barge equipped with two 6 m diameter thrusters. Similar results to Connolly and Crawford (2023) were found in region II and the first part of region III, except that the power ratio was found to be of the order of 70%. The greater power ratio may be explained by the fact that Alwan et al. (2021) used propellers designed for ships (Wageningen B-series screw

propellers (Bernitsas et al., 1981)) for the thrusters, whereas Xu et al. (2021) used thrusters optimized for dynamic positioning. Moreover, Alwan et al. (2021) found that the power ratio re-increases in the second part of region III because of increasing mean drift forces due to waves action. This effect was not observed in (Connolly and Crawford, 2023) because their floater design (a semi-sub) is much more transparent to the waves than the barge used by Alwan et al. (2021).

To date, MUFOWTs energy performance has been investigated in (Tsujimoto et al., 2009; Annan et al., 2023; Connolly and Crawford, 2023). In the latter study, the authors showed that the UFOWT design of Xu et al. (2021) can generate 25% more energy when allowed to move (MUFOWT) than when it is kept stationary (SUFOWT). However, it is uncertain that this energy gain translates in a similar increase in capacity factor. Indeed, as a MUFOWT's average position changes with time, it can be expected that it would have to be relocated from time to time. The required energy consumption and the energy production loss during relocation is currently unknown, but it could be significant. As mentioned in (Connolly and Crawford, 2023), the capacity factor optimization of a MUFOWT would require the implementation of a weather-routing algorithm.

A SUFOWT was also considered in Santarromana et al. (2024). The design consists of a 5, 8, 10 or 15 MW wind turbine mounted on the UMaine Volturn US-S reference platform (which is a semi-submersible platform) equipped with 1 to 30 DP thrusters. It is shown that the percent of annual expected turbine energy output that is consumed by the thrusters can be reduced to less than 20% by increasing the number and diameter of thrusters. Also, that study points out the fact that thrusters power consumption in region IV (above cut-out wind speed) is non-zero; mainly because of drag forces on wind turbine mast and rotor.

From the previous literature, it can be concluded that the energy efficiency of stationary unmoored floating offshore wind turbines depends significantly on:

- The floater design: it should be transparent to the waves to minimize drift forces,
- The DP system design: it should include as many large diameter thrusters as practicable to maximize their thrust generation efficiency.

The rotor design was also identified as a design parameter by Connolly and Crawford (2023) but its effect was not investigated in their study. Moreover, previous studies (Blackford, 1985; Gaunaa et al., 2009) on flow driven vehicles using rotors for energy conversion have shown that “the rotor design for the these kinds of applications is generally different from the rotor design for maximization of power output for a conventional stationary wind turbine which is close to $a = 1/3$ in order to have a high $C_P$. (Gaunaa et al., 2009)”. A similar conclusion was found for the rotors of the water turbines of energy ships (Babarit et al., 2020). Those works indicate that it is worth investigating the effect of rotor design on energy efficiency of SUFOWTs, which is the aim of the present study.

The remainder of this paper is structured as follows. In Section 2, the UFOWT model used in this study is presented. Section 3 focuses on the effect of rotor design on energy performance while the its effect on levelized cost of energy is investigated in Section 4. Conclusions are summarized in Section 5.

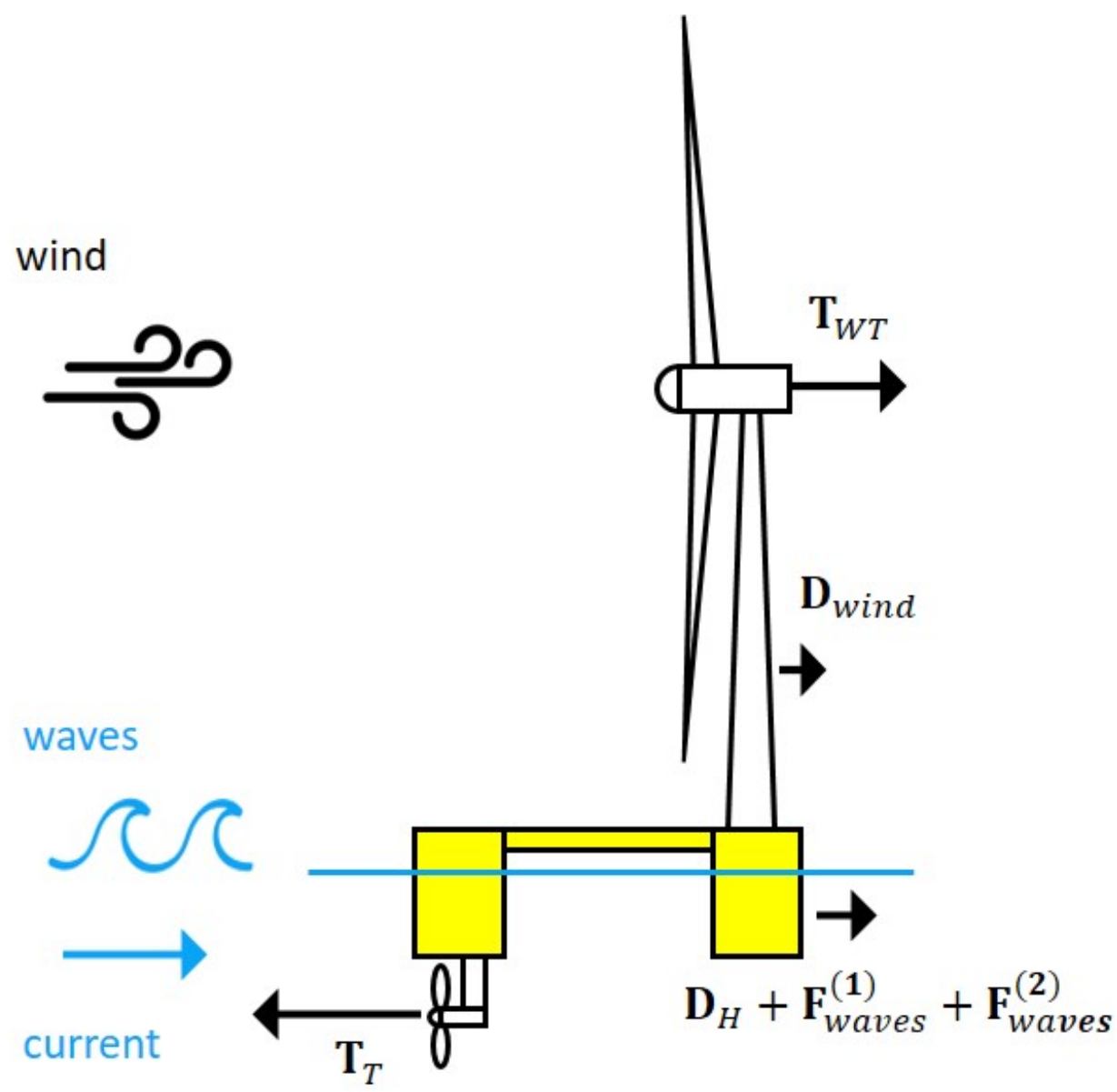

**Figure 2.** External forces applying to an unmoored floating offshore wind turbine

## 2 UFOWT model

In the most general situation, the forces applying to a SUFOWT (Figure 2) are:

- The thrust force generated by the rotor of the wind turbine. It is denoted $\mathbf{T}_{WT}$,
- The aerodynamic drag force due to wind action on the turbine mast and floater superstructure (emerged part of the floater). It is denoted $\mathbf{D}_{wind}$,
- The thrust force generated by the thrusters of the DP system. It is denoted $\mathbf{T}_{T}$,
- A hydrodynamic drag force due to current and floater velocity. It is denoted $\mathbf{D}_{H}$,
- Wave induced hydrodynamic forces due to waves action on the submerged part of the floater: $\mathbf{F}_{waves}$.

The wave-induced force $\mathbf{F}_{waves}$ is usually modelled using potential flow theory as the sum of a first order component $\mathbf{F}^{(1)}_{waves}$ and a second order component $\mathbf{F}^{(2)}_{waves}$:

$$\mathbf{F}_{waves} = \mathbf{F}^{(1)}_{waves} + \mathbf{F}^{(2)}_{waves} \tag{1}$$

The first order component is a time-dependent force whose frequency spectrum is the same as the incident wave. Its time average is zero. The second order component includes a non zero component which is the mean drift force and a time-dependent force whose spectrum includes the sum and difference frequencies of the incident wave spectrum. As shown in (Alwan et al.,

2021), the mean drift force may impact significantly the energy performance of a UFOWT (depending on the platform design and environmental conditions).

The total hydrodynamic force applying to the platform is $\mathbf{D}_H + \mathbf{F}^{(1)}_{waves} + \mathbf{F}^{(2)}_{waves}$ as depicted in Figure 2.

In this study, the aim is at understanding the effect of rotor design on the efficiency of SUFOWTs. Therefore, the following simplifying assumptions are made:

- The DP system is controlled such as first order wave forces, first order platform motion and time-dependent second order wave forces are not compensated, avoiding having to take them into account. This is typically achieved using wave frequency filtering of the position and velocity measurements (Fossen, 2012),
- The hydrodynamic drag force and the drift forces are neglected. In practice, the surface current is about 3% of the wind speed at 10 m height for winds between 5 and 30 m/s (Weber, 1983), thus the force due to current is expected to be small. For the drift forces, it is expected that a highly transparent floater is used (e.g a semi-submersible platform with a reduced water plane area),
- The aerodynamic drag force is neglected, which is quite acceptable in region II and III (Santarromana et al., 2024),
- The wind turbine rotor thrust force and the thrusters thrust force are perfectly aligned,
- The UFOWT pitch motion is neglected,
- The wind is uniform.

Under those assumptions, the thrusters of the DP system have only to counteract the wind turbine rotor thrust force:

$$T_T = T_{WT} \tag{2}$$

### 2.1 Wind turbine model

The wind turbine is modeled using the actuator disc theory (Manwell et al., 2009). Thus, the wind turbine rotor thrust force and the generated power can be written:

$$T_{WT} = \frac{1}{2}\rho_a A W^2 C_T \tag{3}$$

$$P_{WT} = \frac{1}{2}\rho_a A W^3 C_P \eta_{WT} \tag{4}$$

where $\rho_a$ is the air density, $A$ is the turbine swept area, $W$ is the true wind velocity, $C_T = 4a(1-a)$ is the thrust coefficient where $a$ is the induction factor, $C_P = 4a(1-a)^2$ is the theoretical aerodynamic power coefficient from Froude-Rankine actuator disc theory. $\eta_{WT}$ is the wind turbine efficiency defined as the ratio of the electrical power generated by the turbine to the theoretical aerodynamic power. It accounts for aerodynamic losses (finite number of blades, drag, tip losses, etc.), mechanical losses (drivetrain efficiency) and conversion losses (generator efficiency). In this study, we used $\eta_{WT} = 75\%$ which is a conservative estimate for the efficiency of reference offshore turbines (Jonkman et al., 2009; Allen et al., 2020; Bortolotti et al., 2019; Gaertner et al., 2020).

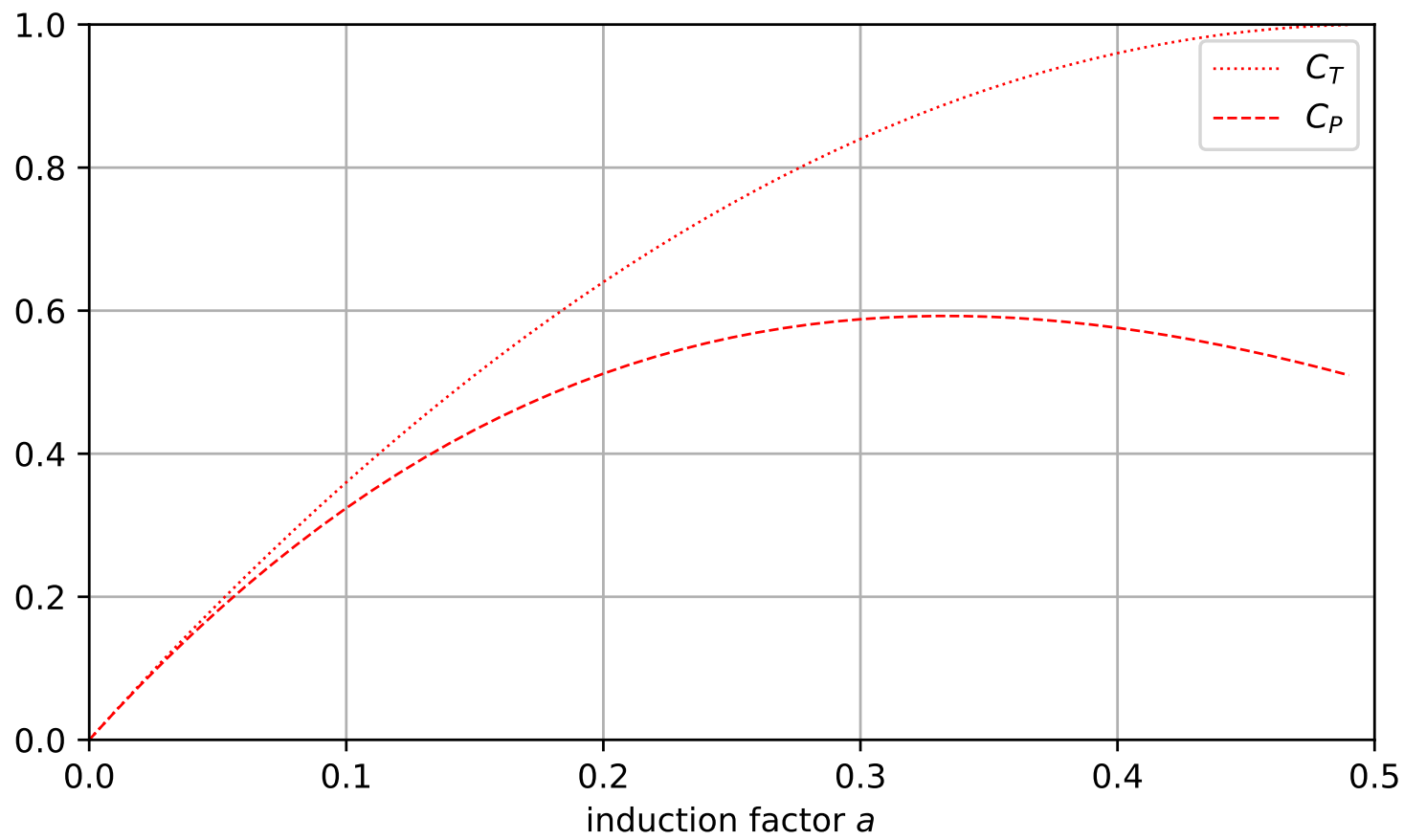

**Figure 3.** Trust and power coefficient as function of induction factor of an ideal wind turbine rotor ($\eta_{WT} = 100\%$)

It is well known that the theoretical maximum power coefficient of a wind turbine is $C_{P,max} = {}^{16}/_{27}$ (Betz's limit). It is achieved for induction factor $a = {}^{1}/_{3}$. Rotors of conventional wind turbines are optimized to get as close as possible to that $C_{P,max}$. However, it comes at cost of high thrust coefficient as can be seen in Figure 3. While it is not a critical challenge when station-keeping does not consume power (as for onshore, bottom-fixed or moored floating offshore wind turbines), the situation is different for UFOWTs as will be shown in what follows.

### 2.2 Thrusters model

The thrusters model follows the approach used in Santarromana et al. (2024). According to ABS (2024), the thrust $T_0$ (in N) of a ducted propeller can be related to its power consumption $P_0$ (in W) using:

$$T_0 = K(P_0 D)^{2/3} \tag{5}$$

where $D$ is the diameter of the thruster propeller (in m) and $K$ is a constant equal to 12.5 kg$^{1/3}$.m$^{-1}$.

Assuming that the SUFOWT is equipped with $N$ identical thrusters and that they all deliver the same thrust, one can show that the thrusters total power consumption $P_T$ is:

$$P_T = \frac{N}{D}\left(\frac{T_T}{KN}\right)^{3/2} \tag{6}$$

eawe european academy of wind energy
Open Access WIND ENERGY SCIENCE DISCUSSIONS

## 3 Energy production

### 3.1 Net power generation and net power coefficient

Equating the wind turbine rotor thrust (Equation (3)) and the thrusters delivered thrust in Equation (6), the thrusters power consumption can be written:

$$P_T = \frac{1}{2}\rho_a A W^3 \times 4a(1-a)^2 \times \sqrt{\frac{2\rho_a A}{K^3 N D^2}\frac{a}{1-a}} \tag{7}$$

in which we recognize the wind generated power divided by the wind turbine efficiency (Equation (4)). Inserting that equation in Equation (7), one can show:

$$P_T = P_{WT} \times \frac{1}{\eta_{WT}}\sqrt{\frac{\pi\rho_a}{2K^3}}\sqrt{\frac{1}{\delta}\frac{a}{1-a}} \tag{8}$$

where $\delta$ is the surface ratio between the swept areas of the thrusters and the wind turbine rotor:

$$\delta = \frac{N\pi D^2}{4A} \tag{9}$$

Note that the term $\frac{1}{\eta_{WT}}\sqrt{\frac{\pi\rho_a}{2K^3}}\sqrt{\frac{1}{\delta}\frac{a}{1-a}}$ in Equation (8) corresponds to the power ratio (that is the ratio of the thrusters power consumption to the wind turbine generated power).

The net power generation $P_{net}$ is the difference between the power generated by the wind turbine and the thrusters power consumption. Thus, combining Equations (4) and (8), the net power generation reads:

$$P_{net} = P_{WT} \times \left(1 - \frac{1}{\eta_{WT}}\sqrt{\frac{\pi\rho_a}{2K^3}}\sqrt{\frac{1}{\delta}\frac{a}{1-a}}\right) \tag{10}$$

This last equation complies with the observations in previous studies that the net power generation increases when reducing the wind turbine diameter/surface area $A$ or increasing the diameter $D$ and/or number of thrusters $N$ (Alwan et al., 2021; Xu et al., 2021; Santarromana et al., 2024). Indeed, both leads to an increase of the surface ratio $\delta$ thus a reduction of power self-consumption in Equation (10).

Now let us consider the net power coefficient $C_{P,net}$. It reads:

$$C_{P,net} = 4a(1-a)^2\eta_{WT} \times \left(1 - \frac{1}{\eta_{WT}}\sqrt{\frac{\pi\rho_a}{2K^3}}\sqrt{\frac{1}{\delta}\frac{a}{1-a}}\right) \tag{11}$$

The maximum of $C_{P,net}$ is not as easy to determine as for a conventional wind turbine. Moreover, it depends on the surface ratio $\delta$ and the wind turbine efficiency $\eta_{WT}$. Table 1 shows the surface ratio corresponding to designs that have been investigated in the literature. One can see that its order of magnitude varies in range 0.001 - 0.1.

Figure 4 shows the net power coefficient and power ratio as function of the induction factor for various surface ratio $\delta$ in range 0.001 to 0.05. One can see that whatever the surface ratio is, the induction factor that maximizes the power coefficient is smaller than $1/3$. For $\delta = 0.05$, the optimal induction factor is approximately 0.31, for which the net power coefficient is

WIND ENERGY SCIENCE DISCUSSIONS
| Reference | $A$ (m$^2$) | $D$ (m) | N | $\delta$ |
|---|---|---|---|---|
| (Alwan et al., 2021) | 4 800 | 6 | 2 | 0.012 |
| (Xu et al., 2021) | 11 000 | 3.6 | 4 | 0.0036 |
| (Santarromana et al., 2024) | 12 000 - 43 000 | 3.8 - 5.5 | 4 - 15 | 0.00075 - 0.03 |

**Table 1.** Characteristics of SUFOWT designs considered in the literature.

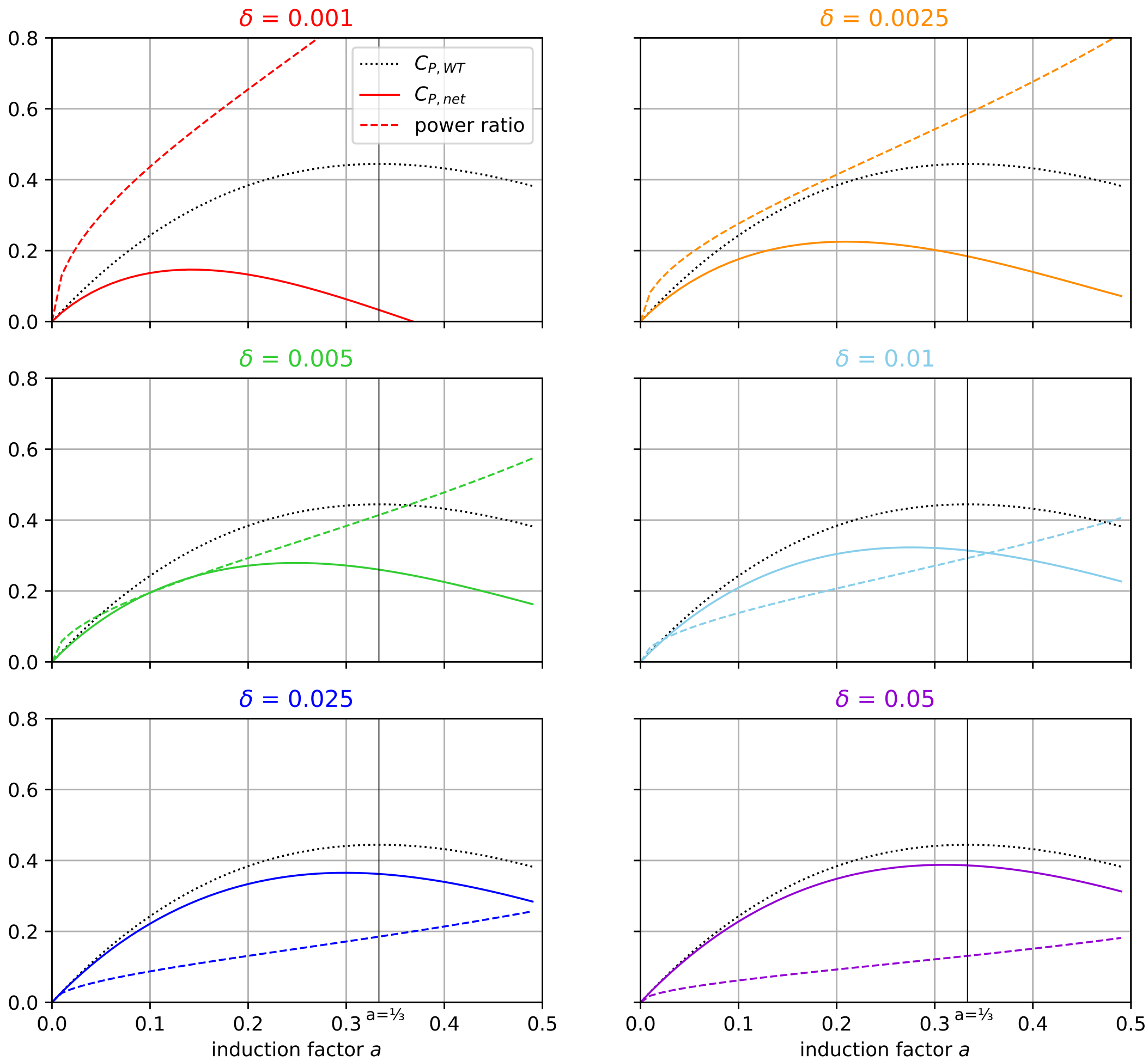

**Figure 4.** Net power coefficient (solid line) and power ratio (dashed line) of a stationary unmoored floating offshore wind turbine as function of induction factor and surface ratio. The dotted black line is the power coefficient of the wind turbine.

approximately 0.388. This makes however little difference with the net power coefficient for $a = 1/3$, which is 0.387 (-0.3%).
155 The difference is more significant for smaller surface ratios. For $\delta = 0.01$, the optimal net power coefficient is 0.323 whereas

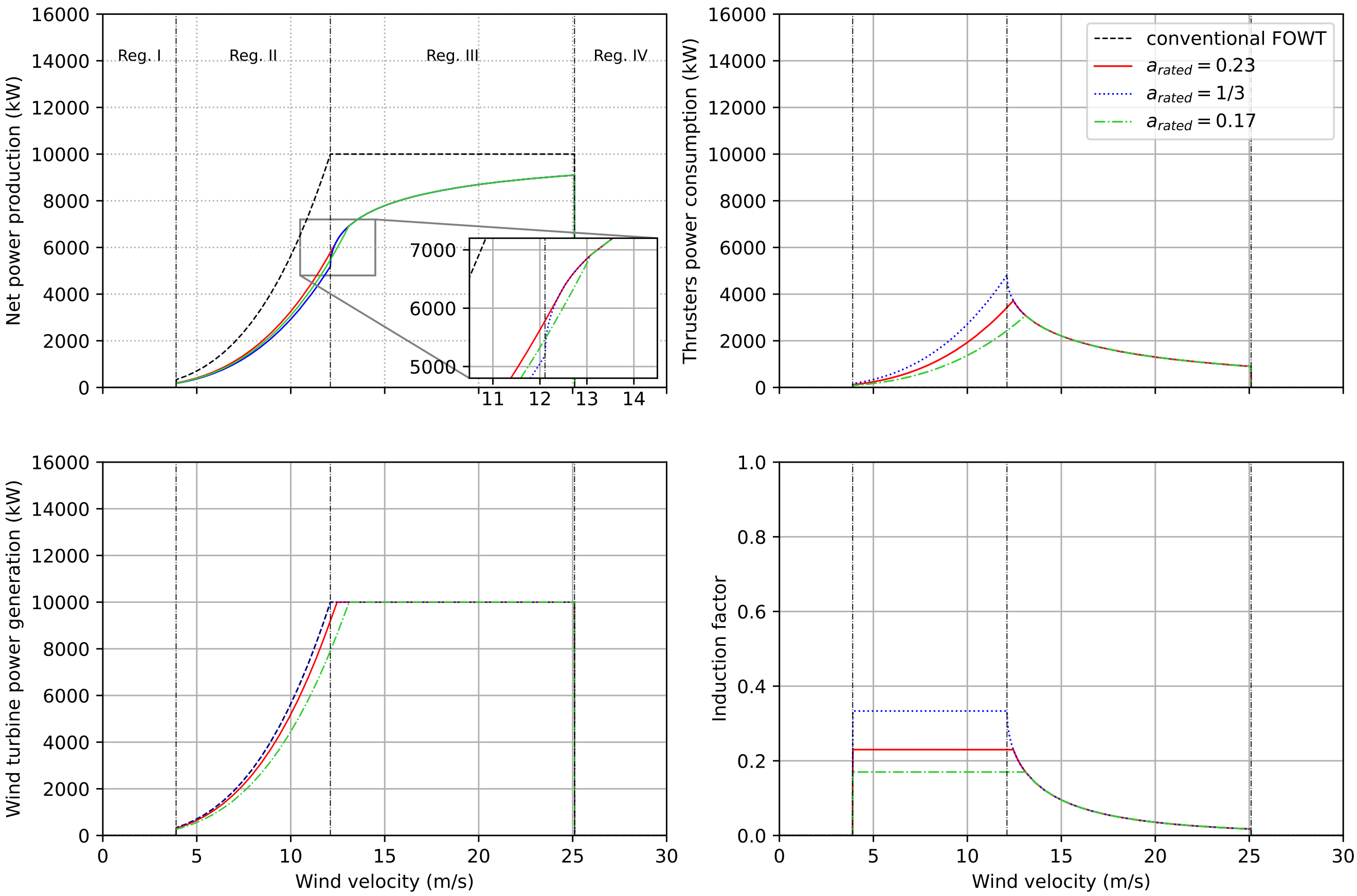

**Figure 5.** Power performance curves of an example SUFOWT as function of wind velocity and rotor design ($a_{rated}$). The regions of operation in the top left corner are for the conventional FOWT.

the net power coefficient at induction factor $a = 1/3$ is 0.315 (-2.5%). For $\delta = 0.0025$, the optimal net power coefficient is reached at $a = 0.21$, and is 21% greater than the power coefficient at induction factor $a = 1/3$.

Figure 4 shows that another advantage of optimizing the induction factor is that the power ratio also reduces. For $\delta = 0.05$, the power ratio for the optimized induction factor is 4.5% smaller than that for $a = 1/3$. For $\delta = 0.01$, it is 11%. For $\delta = 0.0025$, it is 27%. Thus, optimizing the induction factor (thus the wind turbine rotor design) for the SUFOWT application becomes increasingly beneficial for the power ratio as the surface ratio decreases. This is important because the power ratio reflects the power requirement from the thrusters, which relates to their cost.

### 3.2 UFOWT power curves

A wind turbine power curve represents the power that it can produce as function of a stationary hub-height wind speed. For the SUFOWT concept considered in this study, it is given by Equation (10) in which the induction factor varies depending on

WIND ENERGY SCIENCE
DISCUSSIONS

the wind speed. Below the cut-in wind speed $W_{cut-in}$ (Region I), the wind turbine does not operate, thus the induction factor is 0. In Region II (between cut-in wind speed and rated wind speed $W_{rated}$ for which the wind turbine power reaches its rated power $P_{WT,rated}$), the wind turbine operates at rated induction factor $a_{rated}$. Between rated wind speed and the cut-out wind speed $W_{cut-out}$ (Region III), the induction is reduced in order to keep the generated power constant and equal to the rated power. Above the cut-out wind speed $W_{cut-out}$ (Region IV), the wind turbine does not operate, thus the induction factor is 0. The induction factor in Equation (10) is thus taken as follows:

$$a(W) = \begin{cases} 0 & W \leq W_{cut-in} \text{ [Region I]} \\ a_{rated} & \text{if } W_{cut-in} < W \leq W_{rated} \text{ [Region II]} \\ a^*(W) & \text{if } W_{rated} < W \leq W_{cut-out} \text{ [Region III]} \\ 0 & W > W_{cut-out} \text{ [Region IV]} \end{cases} \tag{12}$$

where $W_{rated}$ is such as:

$$2\rho_a A W_{rated}^3 a_{rated}(1-a_{rated})^2 \eta_{WT} = P_{WT,rated} \tag{13}$$

and $a^*(W)$ is the solution of:

$$2\rho_a A W^3 a^*(1-a^*)^2 \eta_{WT} = P_{WT,rated} \tag{14}$$

Note that the rated wind speed depends on $a_{rated}$ as can be seen in Equation (13).

Figure 5 shows examples of power curves calculated using Equation (12). The wind turbine diameter is 164 m corresponding to a rotor disc area $A \approx 21\,000$ m$^2$. The efficiency is approximated as $\eta_{WT} = 75\%$, as explained in Section 2.1. The rated power is $P_{WT,rated} = 10000$ kW, the cut-in wind speed is $W_{cut-in} = 4$ m/s, the cut-out wind speed is $W_{cut-out} = 25$ m/s and the surface ratio is chosen equal to $\delta = 0.0037$, as for the cost-optimal design found in Santarromana et al. (2024). Three rated induction factor $a_{rated}$ are considered: 0.17, 0.23, $1/3$. The corresponding rated wind speeds are: 13.1, 12.4 and 12.1 m/s. $a_{rated} = 0.23$ corresponds approximately to the value that maximizes the net power coefficient according to Equation (11). Results for $a_{rated} = 0.17$ are also plotted for sake of comparison.

As expected, the net power production is the greatest for $a_{rated} = 0.23$ in Region II. However, in Region III, one can see that the net power production increases very rapidly with increasing wind velocity for $a_{rated} = 1/3$, eventually leading to the net power production being the same as for $a_{rated} = 0.23$ as soon as the wind velocity is greater than 12.4 m/s.

For $a_{rated} = 0.17$, one can see that the net power is greater than that for $a_{rated} = 1/3$ for all wind speeds in Region II. However, contrary to $a_{rated} = 0.23$, the net power is smaller than that for $a_{rated} = 1/3$ between 12.1 m/s (rated wind speed for $a_{rated} = 1/3$) and 13.1 m/s (rated wind speed for $a_{rated} = 0.17$). Above 13.1 m/s, the net power is the same.

### 3.3 Net annual expected energy production

The net annual expected energy production $E_{net}$ is estimated according to:

$$E_{net} = 8760 \times \int f(W) \times P_{net}(W) dW \tag{15}$$

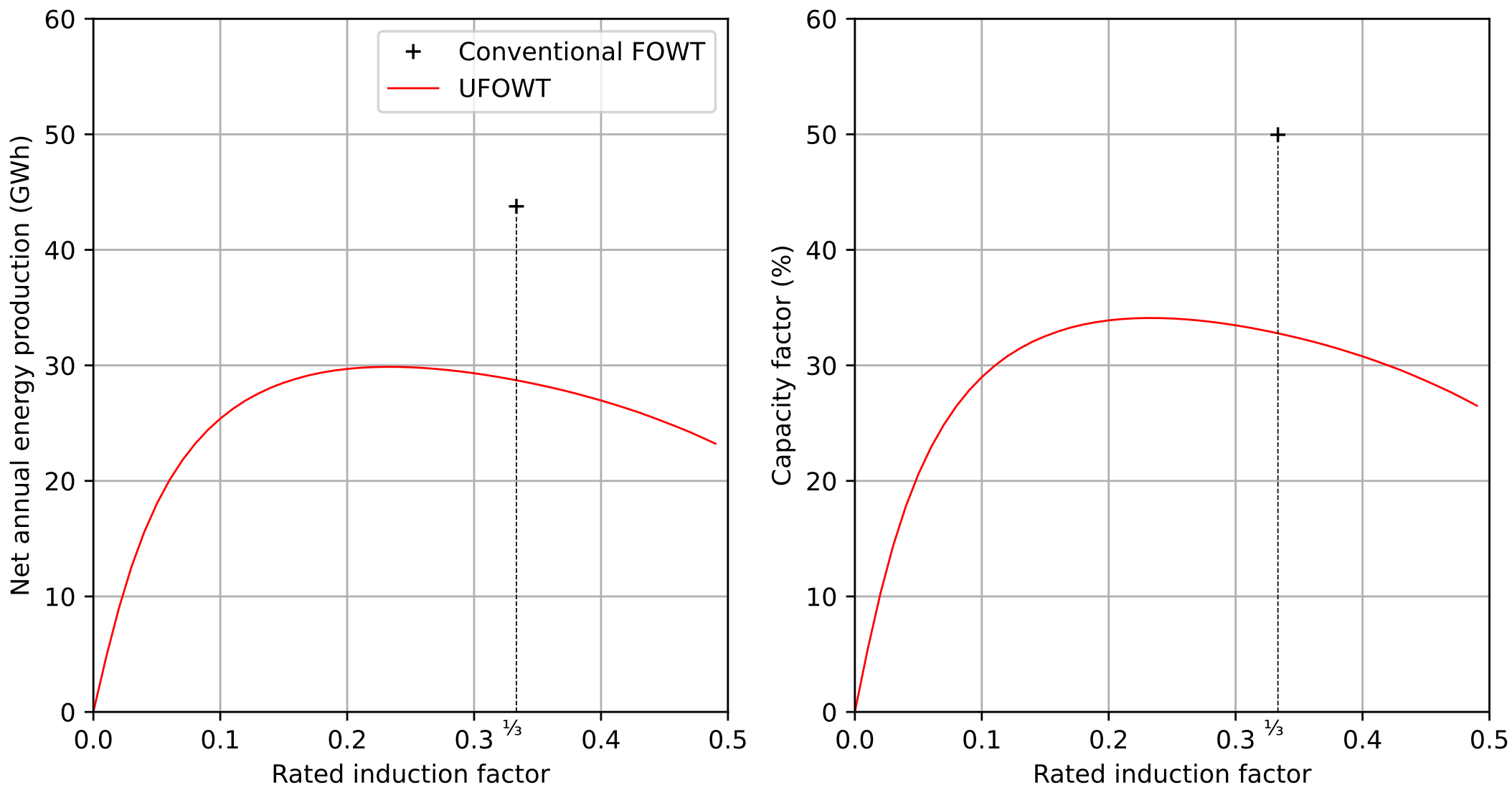

**Figure 6.** Net annual expected energy production and capacity factor as function of the rotor design ($a_{rated}$). The surface ratio is $\delta = 0.0037$.

where we count 8760 hours in a year, $f$ is the wind probability density function and $P_{net}$ is the UFOWT power curve.

In this study, a site with class 8 wind resource is considered (NREL, 2024). The probability density function follows a Weibull distribution. The mean wind speed is 9.41 m/s.

The net capacity factor is derived from the net annual expected energy production using:

$$CF_{net} = \frac{E_{net}}{8760 \times P_{WT,rated}} \tag{16}$$

Figure 6 shows the net annual expected energy production and capacity factor as function of the rated induction factor $a_{rated}$ for the same example SUFOWT as in the previous section. One can show that $a_{rated} = 1/3$ is neither optimal for the annual energy production nor for the capacity factor. Indeed, the maximum annual energy production of 29.9 GWh is obtained for rated induction factor $a_{rated} = 0.23$. In comparison, the annual energy production at $a_{rated} = 1/3$ is 28.8 GWh (-3.7%). Note that the annual energy production is 32% smaller than that of a conventional FOWT based on the same wind turbine as the SUFOWT (43.8 GWh).

The SUFOWT maximum capacity factor is 34.1%. It is reached for $a_{rated} = 0.23$. In comparison, the capacity factor for $a_{rated} = 1/3$ is 32.9% (-3.5%). That of its conventional FOWT counterpart is 50.0% (+46.7%).

## 4 Cost optimization

Figure 4 shows that in theory it is possible for a SUFOWT to achieve power performance close to that of a conventional FOWT provided that the surface ratio is small enough. However, minimizing the surface ratio does not necessarily lead to the lowest Levelized Cost Of Energy (LCOE) as was shown in Santarromana et al. (2024). Indeed, in their study, they optimized

| | | | | | |
|---|---|---|---|---|---|
| Turbine rated power (MW) | $P_{WT,rated}$ | 5 | 8 | 10 | 15 |
| Rotor diameter (m) | - | 126 | 164 | 198 | 240 |
| Turbine specific power (W/m$^2$) | - | 400 | 379 | 325 | 332 |
| Turbine efficiency (%) | $\eta_{WT}$ | 75 | 75 | 75 | 75 |
| Cut-in wind speed (m/s) | $W_{cut-int}$ | 3 | 3 | 3 | 3 |
| Cut-out wind speed (m/s) | $W_{cut-out}$ | 25 | 25 | 25 | 25 |

**Table 2.** Wind turbines characteristics

the number of thrusters, their diameter and the wind turbine size with respect to LCOE. Their best configuration is an 8 MW wind turbine with four 5 m diameter thrusters. For that configuration, they estimated the LCOE aboard the wind turbine to be approximately 30 USD/MWh greater than that of a conventional FOWT.

The optimal design of Santarromana et al. (2024) corresponds to surface ratio $\delta = 0.0037$, whereas their optimization range encompassed surface ratio from 0.00075 to 0.03. For surface ratio $\delta = 0.0037$, Figure 6 shows that optimizing the rotor design to a chosen rated induction factor can both increase the net energy production and the capacity factor. Thus, the effect of rotor design optimization on LCOE is investigated in what follows.

The approach is inspired by Santarromana et al. (2024), *i.e.* we consider:

- Four possible offshore wind turbines. Their dimensions correspond to that of the NREL 5 MW reference wind turbine (Jonkman et al., 2009), the 8 MW reference wind turbine of Desmond et al. (2016), the IEA 10 MW reference wind turbine (Bortolotti et al., 2019), and the IEA 15 MW reference wind turbine (Gaertner et al., 2020). The cut-in wind speed and cut-out wind speeds were taken equal to 3 m/s and 25 m/s respectively. Their characteristics are summarized in Table 2. Note that in contrast to Santarromana et al. (2024), we used Equation (3) for the wind turbine thrust and Equation (4) for the wind turbine generated power whereas they used data specific to the aforementioned reference wind turbines,

- A number of thrusters whose specifications are taken from Table 4 in Santarromana et al. (2024). They are recalled in Table 3.

As mentioned previously, it is assumed that the turbine and thrusters are mounted on a platform highly transparent to the waves (e.g a semi-submersible platform with a reduced water plane area), in order to minimize drift forces.

The optimization was performed using brute-force: for each wind turbine and rated induction factor, each thruster in Table 3 was considered in number from 1 to 30 and the configuration with the least cost of energy was retained.

eawe european academy of wind energy
Open Access WIND ENERGY SCIENCE DISCUSSIONS

| Nominal input power (kW) $P_{0,nominal}$ | Propeller diameter (m) $D$ | Nominal thrust (kN) $T_{0,nominal}$ | Unit price (USD) $CAPEX_{thruster}$ |
|---|---|---|---|
| 3200 | 3.2 | 598 | 995100 |
| 4000 | 3.5 | 726 | 1070000 |
| 4800 | 3.8 | 866 | 1230500 |
| 5500 | 4.1 | 998 | 1534380 |
| 6500 | 4.5 | 1187 | 2210620 |
| 3200 | 4.1 | 695 | 1177000 |
| 3800 | 4.5 | 830 | 1444500 |
| 4500 | 5.0 | 996 | 1765500 |
| 5500 | 5.5 | 1214 | 2461000 |

**Table 3.** Thrusters characteristics.

### 4.1 Cost model

The LCOE calculation follows the method proposed in Santarromana et al. (2024), which is an adaptation to the method described in the NREL Annual Technology Baseline (ATB) (NREL, 2024):

$$LCOE = \frac{CRF \times PFF \times CFF \times OCC_{net} + FOM}{E_{net}} \tag{17}$$

where $CRF$ is the Capital Recovery Factor, $PFF$ is the Production Finance, $CFF$ is the Construction Finance Factor, $OCC_{net}$ is the Overnight Capital Cost, $FOM$ is the Fixed Operations and Maintenance annual cost.

Santarromana et al. (2024) used the 2023 NREL ATB database (NREL, 2023) whereas we used the updated 2024 database (NREL, 2024) in the present study. In this respect, $CRF = 5.8\%$, $PFF = 1.056$ and $CFF = 1.109$. The $FOM$ excluding the thrusters is in range 61 - 74 USD/kW. The $FOM$ of the thrusters was estimated by Santarromana et al. (2024) to 63000 USD per thruster.

The $OCC_{net}$ in USD is taken equal to:

$$OCC_{net} = CAPEX_{FOWT} + N \times CAPEX_{thruster} \tag{18}$$

where $CAPEX_{FOWT}$ is the wind turbine plus floater cost, $CAPEX_{thruster}$ is the cost of each thruster. We recall that $N$ is the number of thrusters.

The cost of thrusters $CAPEX_{thruster}$ is taken according to data provided in Table 4 in Santarromana et al. (2024). For the wind turbine plus floater cost $CAPEX_{FOWT}$, according to NREL (2024), the 2030 cost of floating offshore wind turbines is in range 4000 - 9000 USD/kW for a site of wind speed class 8. This figure includes mooring cost which we assume to account for 11% of the total floating offshore wind turbine cost following Santarromana et al. (2024). Therefore, we used 3600 - 8000 USD/kW for $CAPEX_{FOWT}$ in Equation (18).

eawe european academy of wind energy
Open Access WIND ENERGY SCIENCE DISCUSSIONS

### 4.2 Results

Figure 7 shows the optimization results. Left plots show the results when using the lowest costs of the costs ranges. Right plots show the results when using the highest costs. From top to bottom, Figure 7 shows the LCOE as function of the rated induction factor, the capacity factor, the optimized thrusters diameter, the optimized number of thrusters, the corresponding surface ratio (Equation (9)) and the thrusters use ratio (see below for the definition of the thrusters use ratio).

In the top graphs, one can see that the optimal LCOE is obtained for induction factors in range 0.22 to 0.25. In comparison to the LCOE for induction factor equal to $1/3$, the LCOE is reduced by 2.5 to 4.3%.

Results show that the LCOE also depends on the rated power. The smallest LCOE is obtained for rated power equal to 10 MW [115 - 205 USD/MWh], then 15 MW [115 - 206 USD/MWh], then 8 MW [120 - 215 USD/MWh] and finally 5 MW [122 - 219 USD/MWh]. This ranking corresponds to that for the capacity factor (39.1 - 42.0% for 10 MW rated power, 38.8 - 41.6% for 15 MW, 37.3 - 40.1% for 8 MW, 37.4 - 39.7% for 5 MW), which itself corresponds to the wind turbines specific power ranking (rated power per unit rotor swept area, see Table 2). This indicates that lower specific power turbines could further decrease the LCOE of SUFOWTs.

In comparison, the LCOE of conventional FOWTs (dashed lined in Figure 7) are 72 - 149 USD/MWh for 5 MW rated power, 70 - 146 USD/MWh for 8 MW, 66 - 137 USD/MWh for 10 MW and 67 - 138 USD/MWh for 15 MW. We recall that it is the cost of energy onboard the platform (it does not take into account grid connection cost). Thus, the LCOE of SUFOWTs is 50 to 70% greater than their conventional FOWTs counterparts. This is mainly due to the capacity factor of SUFOWTs being approximately 30 % smaller than their conventional FOWTs counterparts.

The optimized surface ratio $\delta$ displays limited variability as function of rated power. For optimal induction factor, it is of the order of 0.003 for the low cost assumptions. It is of the order of 0.004 for the high cost assumptions. Results show that the optimized surface ratio is obtained by increasing the number of thrusters when increasing the size of the wind turbine, the optimized thrusters diameter being the same for all four wind turbines (5 m).

The bottom plots in Figure 7 shows the thrusters use ratio, which is defined as the ratio of the maximum thrusters power (that is the thrusters power at rated wind speed) to the thrusters nominal input power as given in Table 3. One can see that the thrusters use ratio is low, of the order of 10 to 20% depending on the cost assumptions. This shows that the thrusters considered in this study - which are designed for dynamic positioning of ships - are oversized for the SUFOWT application. Thus, thrusters designs dedicated to the SUFOWT application could help reduce costs.

## 5 Conclusions

In this study, we investigated the effect of wind turbine rotor design on the energy performance and cost of energy of a stationary unmoored floating offshore wind turbine (SUFOWT). The wind turbine rotor design is represented by its rated induction factor.

Mathematical relationships for energy performance were developed. They show that a key parameter of SUFOWT energy performance is the surface ratio (ratio of the thrusters propellers swept area to the turbine swept area), and that the energy performance of SUFOWTs tends to that of conventional FOWTs when the surface ratio is large. However, large surface ratio

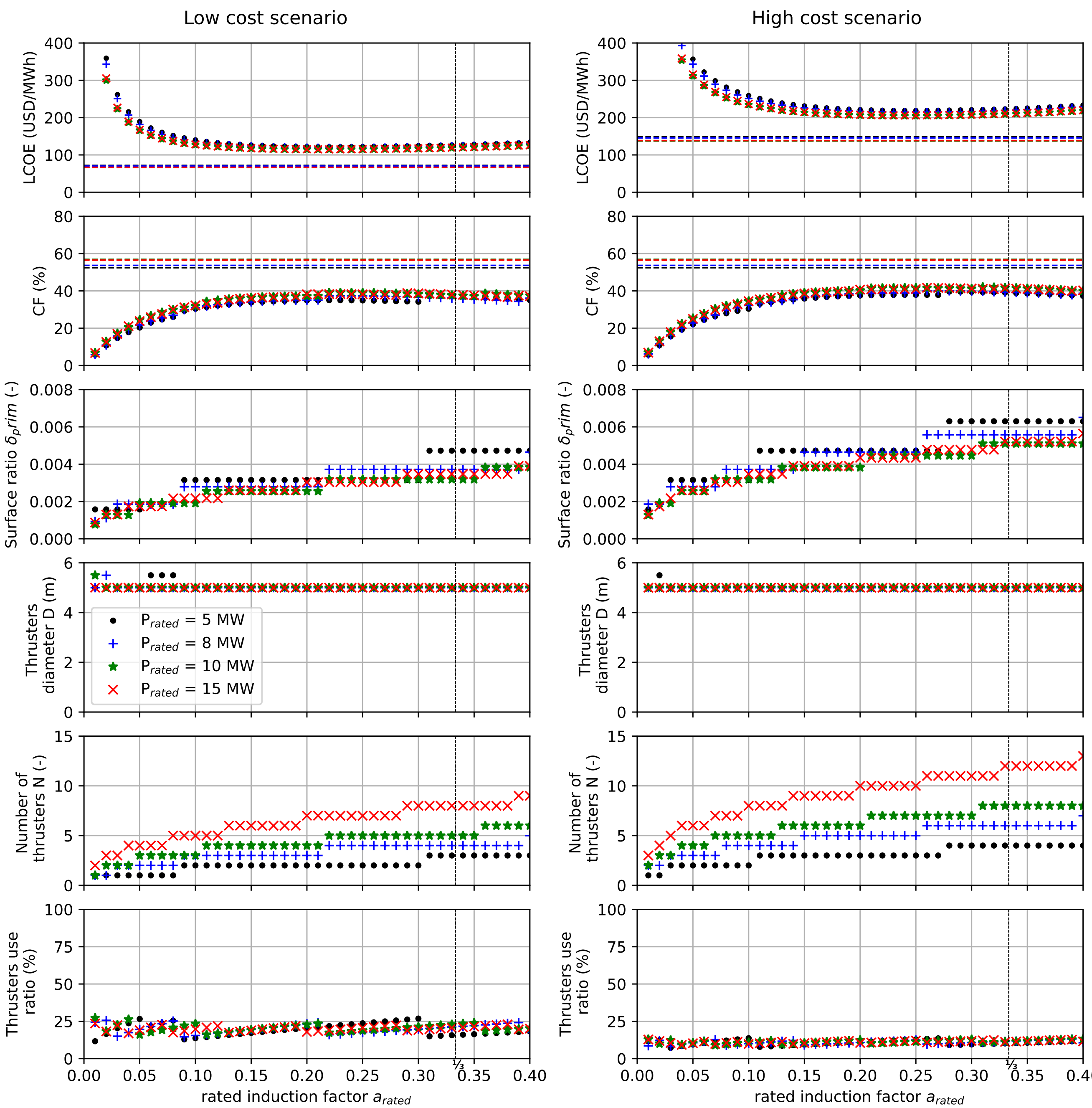

**Figure 7.** Results of cost of energy optimization. Left plots correspond to the low cost scenario (right plots correspond to the high cost scenario). Symbols correspond to results for SUFOWTs. Dashed lines are results for conventional FOWTs.

requires large number of thrusters and/or thrusters with large diameter, which can lead to a sub-optimal SUFOWT design from the cost of energy perspective as was shown by Santarromana et al. (2024). For realistic surface ratios, our results show that the induction factor which maximizes the net power coefficient (taking into account the thrusters power consumption) is significantly smaller than the value of $1/3$ which maximizes the wind turbine power coefficient.

Cost of energy optimization was also performed. Results show that the optimal induction factor of SUFOWTs is also smaller than that of conventional UFOWTs (in range 0.22 to 0.25 for the designs considered herein). Thus, wind turbine rotors designed for SUFOWTs should be developed for cost optimization. However, results show that the cost of energy reduction is somehow limited, of the order of 2.5 to 4.3% for the considered designs.

Our results also indicate that there two other possible pathways for SUFOWT further cost optimization. They are the turbine specific power and the thrusters design. Indeed, results indicate that cost of energy reduces with decreasing specific power, and that conventional thrusters (used for dynamic positioning of ships) are oversized. The potential of cost reduction of these pathways will be investigated in future work.

*Code and data availability.* Code and data will be made available on request

*Author contributions.* Aurélien Babarit: Conceptualization, Methodology, Software, Investigation, Formal analysis, Writing - Original Draft, Writing - Review & Editing, Visualization, Funding acquisition. Maximilien André: Methodology, Writing - Review & Editing. Vincent Leroy: Methodology, Writing - Review & Editing.

*Competing interests.* The authors declare they have no competing interest.

*Acknowledgements.* This work was supported by Région Pays de la Loire.

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
