# Peer review of "Effect of Rotor Induction and Peak Shaving on Energy Performance and Cost of Stationary Unmoored Floating Offshore Wind Turbines"

_Wind Energy Science, 2025_

## Author Comment (AC1)

3. Section 3.1: The derivation of Eqt. (7) is not explained in enough detail. It is unclear how this equation for the turbine power is derived by equating the wind turbine thrust to the thrust of the dynamic positioning thrusters.

The details of the derivation are as follows:

Eq. 3:   $T_{\mathrm{WT}} = \frac{1}{2}\rho_a A W^2 C_T$

Eq. 6:   $P_{\mathrm{T}} = \frac{N}{D}\left(\frac{T_{\mathrm{T}}}{KN}\right)^{3/2}$

Equating the wind turbine rotor thrust (Eq. 3) and the thrusters delivered thrust in Eq. 6 (line 129 of the manuscript):

$$P_{\mathrm{T}} = \frac{N}{D}\left(\frac{\frac{1}{2}\rho_a A W^2 C_T}{KN}\right)^{3/2}$$

$$P_{\mathrm{T}} = \frac{N}{D}\frac{\frac{1}{2}\rho_a A W^2 C_T}{KN}\sqrt{\frac{\frac{1}{2}\rho_a A W^2 C_T}{KN}}$$

$$P_{\mathrm{T}} = \frac{1}{KD}\times\frac{1}{2}\rho_a A W^2 C_T \times W\sqrt{\frac{1}{2}\frac{\rho_a A C_T}{KN}}$$

$$P_{\mathrm{T}} = \frac{1}{2}\rho_a A W^3 \times C_T \times \frac{1}{KD}\sqrt{\frac{1}{2}\frac{\rho_a A}{KN}C_T}$$

$$P_{\mathrm{T}} = \frac{1}{2}\rho_a A W^3 \times C_T \times \sqrt{\frac{1}{2}\frac{\rho_a A W^2}{K^3 D^2 N}C_T}$$

Using $C_T = 4a(1-a)$:

$$P_{\mathrm{T}} = \frac{1}{2}\rho_a A W^3 \times 4a(1-a) \times \sqrt{\frac{1}{2}\frac{\rho_a A W^2}{K^3 D^2 N}4a(1-a)}$$

$$P_{\mathrm{T}} = \frac{1}{2}\rho_a A W^3 \times 4a(1-a)^2 \times \sqrt{2\frac{\rho_a A W^2}{K^3 D^2 N}\frac{a}{1-a}}$$

Which corresponds to Eq. 7

---

## Author Response (AR1)

**Authors' response to referee's and public comments.**

**Referee #1**

The paper deals with stationery unmoored floating wind turbines. This topic is very relevant to wind turbines deployed in very deep waters far offshore where mooring costs would be high. The paper initially presents a very good overview of relevant literature. It then applies simple actuator disc theory for wind turbines to evaluate the impact of rotor design on the energy yield and cost of such turbines. The quality of the writing is overall very good and grammatically mistakes are very limited. The presentation of graphs and tables is also very clear. However the following major comments are being brought forward:

**Thank you for the feedback and comments. Please find below our replies.**

1. Section 2: The study ignores the hydrodynamic loads of the waves when sizing the thrusters. This assumption is unrealistic, considering that at high wind speeds the wave induced loads on the floating platform are often larger in magnitude than the wind turbine thrust.

We agree that wave induced loads can be greater than wind turbine thrust. However, we disagree that it is an unrealistic assumption to neglect wave loads when sizing the thrusters because their effect does not need to be compensated by the thrusters (line 93 of the manuscript).

- (i) Indeed, the effect of first order wave loads is essentially platform motions at frequencies equal to that of the waves and most importantly whose time averages are zero (which is why the thrusters do not have to compensate those first order loads),
- (ii) The drift forces (second order forces) are small when using a sufficiently transparent floater (e.g a semi-submersible platform with a reduced water plane area), (line 98 of the manuscript).

In practice, (i) may be achieved by filtering out wave-frequency motions in the thrusters controller. This approach was successfully implemented in [Xu et al., 2021] as can be seen in Figures 1 and 2 below. Figure 1 shows the wind turbine thrust force. Figure 2 shows the thrusters force. One can see that after the initial transient, the thrusters force is relatively stable and very close to minus the wind turbine thrust force.

Figure 2. thrusters force as function of time. The picture is extracted from Figure 15 in [Xu et al., 2021].

Regarding (ii), we have run computations of wind loads and drift forces for the NREL 5MW wind turbine mounted on the OC4 semi-submersible platform.

- The wind loads are computed using Ct table from the NREL 5MW reference article,
- The wave loads are computed using the 2nd order QTF for the OC4 semi-submersible platform. The QTF were computed using the software Nemoh. The wave spectrum is the Pierson-Moskowitz spectrum.

Figure 3 shows the drift force (called wave load in the figure), wind loads (rotor+tower), total load and the ratio of the drift force to the total load. The results show that the wave drift force is less than 10% of the total load in most of the operational region (max 15% percent for the highest wind speed). Moreover, the OC4 semi-submersible is quite oversized for the 5MW turbine. Therefore, this ratio is expected to further reduce as the turbine diameter grows and as the floater gets thinner.